# Effect of Bio-Oils and Wastewater Sludge on the Performance of Binders and Hot Mix Asphalt with High Reclaimed Asphalt Pavement Content

**DOI:** 10.3390/ma17174276

**Published:** 2024-08-29

**Authors:** Robeam S. Melaku, Jun Liu, Daba S. Gedafa

**Affiliations:** 1Addis Ababa Institute of Technology, Department of Civil Engineering, Addis Ababa 365, Ethiopia; robeam.solomon@aait.edu.et; 2Department of Civil Engineering, University of North Dakota, Grand Forks, ND 58202, USA; liujun050612@gmail.com

**Keywords:** waste cooking oil, wastewater sludge, RAP, binder performance, HMA performance

## Abstract

Waste Cooking Oil (WCO), Soy Oil (SO), and Wastewater Sludge (WWS) have great potential to increase reclaimed asphalt pavement (RAP) content for economic and environmental benefits. This study explored the effects of SO and WCO on rutting, fatigue cracking, and low-temperature cracking performance of binders and Hot Mix Asphalt (HMA) with high RAP content. The potential effect of WWS on the performance and compaction efforts of high RAP content mixes at a 10 °C (50 °F) lower compaction temperature than the control compaction temperature was also investigated. The results indicated that 85% of the RAP binders can be incorporated while maintaining similar performance compared to the control by using 15% SO or 12.5% WCO as a rejuvenator with 2.5% virgin binder. Adding 1% WWS by weight of the total binder improved the binder’s rheological properties, the mix’s cracking performance, and the mix’s density at lower compaction temperatures.

## 1. Introduction

Asphaltic material is used to surface about 96% of the United States’ paved roads [1]. The most recycled material in the United States, RAP, is used to construct asphalt pavement to reduce construction costs and environmental impact [2]. During the 2017 construction season, RAP use was expected to decrease the requirement for more than 72 million tons of aggregate and 3.8 million tons (21.5 million barrels) of asphalt binder, with a total estimated value of more than $2.1 billion [1]. About 95% of RAP was recycled into new asphalt mixes in 2021 in the United States [3]. The RAP usage has significant advantages for most state transportation departments. The cracking resistance of mixes with up to 20% RAP can be equal to or better than mixes without RAP [4,5,6,7,8]. However, its use is limited to 25–40% in general. The increasing stiffness of asphalt mixtures with high percentages of RAP, triggered by aged asphalt binder, is why greater volumes are not used in HMA. Increasing the RAP percentage renders the HMA susceptible to fatigue and thermal cracking [9]. Adding softer binders, recycling agents, and warm mix additives are some of the approaches suggested by researchers to reduce stiffness and improve the mixture’s cracking performance with a high RAP content [10,11].

Full or partial replacement of virgin binder by bio-oils in mixes containing RAP could be sustainable if it maintains or improves the performance of high RAP mixes [12]. Saha et al. [13] demonstrated that the addition of 5% WCO or 5% SO with 70% RAP binder and 25% PG 64–28 binder displayed better rutting, fatigue cracking, and low-temperature cracking performance than the control, virgin PG 64–28 mix.

Wen and Bhusal [14] reported that 10% and 30% of WCO-modified PG 82–16 binder improved thermal cracking while reducing the fatigue cracking performance of the base binder and HMA. Zargar et al. [15] reported that WCO’s effect as a rejuvenating agent for an aged binder reduced stiffness. Adding WCO to the RAP binder decreased the ratio of asphaltenes to maltenes since WCO contains additional maltenes [16].

Small dosages of soybean acidulated soap stock reduced the aged binder’s stiffness, increased workability, and improved low-temperature performance [17]. Elkashef et al. [18] studied the effect of adding 6% and 12% SO to the PG 58–28 virgin binder to rejuvenate the 100% RAP mixture. The results indicated that the soybean-derived rejuvenator significantly impacted the RAP binder’s low and high-temperature properties. The HMA results suggested that the SO and virgin binder rejuvenated mixture improved the low-temperature fracture energy more than the mix with virgin binder alone. This study lacked information on comparing low-temperature cracking performance between SO rejuvenated RAP binder and virgin binder mixtures.

The WWS contains a silicate of alkali metal and alkali metal oxides, a potential source for artificial syntactic Zeolite-like chemicals [19,20]. Zeolites, a term referring to a group of minerals composed of hydrated aluminum silicates with alkali metals and alkaline earth metals, are commonly used as warm mix additives [21,22]. For most zeolite-based WMA, the average dosage ranges from 1–2% by weight of the total bitumen content [23,24].

The study completed by Melaku and Gedafa [25] reported that modifying 40% RAP mixes by 1.5% WWS improved the fatigue and low-temperature cracking performance to a greater extent than the control, virgin PG-28 binder mix.

While high RAP in HMA promotes sustainable development, its inclusion in high volumes may negatively affect HMA performance. Mixes with a high RAP percentage are susceptible to fatigue and low-temperature cracking due to the aged binder’s stiffness [4,26]. Mixtures with high RAP content may benefit from adding WWS to enhance their fatigue cracking performance. By lowering the compaction temperature of a high RAP mix HMA, WWS can potentially function as a compaction assist ingredient [25]. Using bio-oils, such as SO and WCO, with a high RAP mix resulted in improved low-temperature cracking performance while indicating a reduction in fatigue cracking performance. The effect of WWS on the performance of SO and WCO-modified high RAP content HMA at a 10 °C (50 °F) lower compaction temperature than HMA with a virgin binder was investigated to determine the potential usage of WWS to improve the rutting, low-temperature performance, and compaction effort.

The objectives of the study were to determine the optimum percentage of RAP, bio-oils such as SO and WCO, WWS, and virgin binder to produce high-performing HMA compared to the control (virgin binder-based mixes); improve the performance of high-percentage RAP HMA using optimum bio-oils such as SO and WCO and a virgin binder; and explore the effect of WWS on low-temperature cracking, fatigue cracking, and compaction effort of the bio-oil modified high RAP mixes.

## 2. Materials and Methods

### 2.1. Experimental Plan

Figure 1 illustrates the experimental plan. It consists of the following three phases: (1) modifying the RAP binder with bio-oils to determine the rheology. The optimum dosages of bio-oils to reduce the stiffness of the RAP binder to a comparable level to the control (PG 58–28) binder were determined. Rutting, fatigue cracking, and low-temperature binder tests were conducted using a dynamic shear rheometer (DSR) Smart Pave 102 supplied by Anton Par Austria, (2) preparing and testing HMA with bio-oils. The rutting, fatigue cracking, and low-temperature cracking performance of HMA with the selected dosages of bio-oils from phase 1 were measured; and (3) exploring the effect of WWS on the bio-oil modified HMA at lower compaction temperatures. In this phase, the impact of 1%WWS on the bio-oil modified high RAP HMA on rutting, fatigue cracking, low-temperature cracking, and the effect on the compaction effect was determined.

### 2.2. Material Selection

The RAP and virgin aggregates were collected from the North Dakota Highway-32 project. The PG 58–28 binder was used as a control binder since it is commonly used in North Dakota and was the grade of the RAP binder before aging. The extracted RAP binder was modified using the two bio-oils, SO from a commercial provider and WCO from the University of North Dakota dining center. The WWS was collected from the Grand Forks Wastewater Treatment Plant. Table 1 shows the characteristics of the two bio-oils and WWS.

### 2.3. Binder Test Design and Effect on HMA Mixes

The rheology of the 100% RAP binder was tested first, followed by modifying the RAP binder by adding WCO and SO. Trial dosages of the bio-oils were added to the Rolling Thin Film Oven (RTFO) aged binder to determine the optimum bio-oil and binder combinations. The RTFO samples represented the binder’s service temperature after compaction or short-term aging. Rheological tests were completed on unaged, RTFO, and Pressure Aged Vessel (PAV) aged samples using a DSR Smart Pave 102 supplied by Anton Paar, Graz, Austria. The optimum percentage of the two bio-oils was 15% of the total binder content by weight. This percentage met all the requirements for rheological properties except for the unaged high-temperature performance at 58 °C, which was less than the AASHTO M320 specifications [27]. The replacement of the two bio-oils with the virgin PG 58–28 binder was completed on the two modified binders (15%SO_85%RAP and 15%WCO_85%RAP) to meet this specification. The replacement of the bio-oil content with the virgin PG 58–28 resulted in 2.5% virgin binder and 12.5% bio-oils, which satisfied all rheological AASHTO M320 specifications [27] while maintaining the high-temperature continuous performance grade (PG) of the RAP binder at 58 °C.

Multiple stress creep recovery (MSCR) tests were conducted using a 25 mm plate for rutting performance. Linear Amplitude Sweep (LAS) tests were used for fatigue performance using an 8 mm plate. The 4 mm parallel plate geometry tests were utilized for low-temperature cracking performance. The tests were performed on the control and all selected bio-oil and bio-oil-modified and virgin binders following the AASHTO 320 specification [27]. The combinations included the virgin as a control (V), 15%SO_85%RAP (15SO), 15%WCO_85%RAP (15WCO), 12.5%SO_2.5Virgin_85%RAP (12.5SO_2.5V), 12.5%WCO_2.5Virgin_85%RAP (12.5WCO_2.5V), and 100%RAP (100RAP) binders.

The HMA mix was prepared using the selected bio-oil-modified binders and the virgin binder. The bio-oil modified mixes of 15% SO and WCO were considered for the mix analysis to determine the effects of failing and unaged AASHTO M320 specification [27] binder test results on the mix result. The impact of WWS on the cracking and compaction effect was investigated on bio-oil modified mixes at a 10 °C (50 °F) lower compaction temperature than the control HMA. Three samples were tested for all binder and mix performance analysis. All binder and mix tests conducted are summarized in Table 2.

### 2.4. RAP Binder Extraction and Binder Testing

The RAP binder was extracted jointly according to the ASTM D 2172/D 2172M standard [28] and the ASTM D 1856 procedure [29]. EnSolv-EX, an n-Propyl Bromide (nPB) solvent from Enviro Tech International, Inc USA, was used in the extraction process. The extracted RAP binder was recovered by the Abson Method following the ASTM D 1856-09 standard [29].

Multiple Stress Creep Recovery (MSCR): The AASHTO T 350 [30] and AASHTO M 332 [31] criteria were followed for the MSCR tests using a DSR. The test is a blind binder modifier. The primary measures used to characterize the binder’s rutting performance are the percent recovery (%R) and the non-recoverable creep compliance at 3.2 Kpa (Jnr_3.2). The maximum threshold values for ordinary, heavy, very heavy, and extreme traffic are 4, 2, 1, and 0.5 for Jnr_3.2, respectively. A lower Jnr value indicates a stiffer binder [32].

Linear Amplitude Sweep (LAS): The LAS is an oscillatory strain sweep test for evaluating asphalt binder fatigue damage that uses linearly increasing load amplitudes according to AASHTO TP 101 [33]. The linear amplitude strain sweep test is used in the LAS test to get data on the damage characteristics of the binder. In contrast, the frequency sweep is used to gather information about the undamaged material properties. The PAV-aged binders were used for the test, conducted at a moderate temperature of 25 °C. A reduction of 35% in the modules is set as a failure criterion in the LAS test. The analysis was completed based on fatigue law parameters A and B, which are the model coefficients that depend on the material characteristics [34]. Fatigue-resistant binders tend to have higher A values and lower absolute B values. The number of cycles for failure calculations using Equation (1) was employed to compare the fatigue cracking performance between the binders.
(1)$$N_{f}=A({\gamma_{max})}^{B}$$

Binder fatigue parameters (N*_f_*) can be adjusted to account for the differences in pavement structure by changing the maximum shear strain (γ*_max_*). Higher strain values may correspond to thinner pavements or heavier traffic loading, while lower strain values may correspond to thicker pavements or lighter traffic loads. The applied binder strain was estimated using the strain in the pavement layer multiplied by 50. Binder fatigue parameters (N*_f_*) were determined at 2.5% and 5% binder strain levels, corresponding to 500 and 1000 microstrains in the pavement layer, respectively [34].

Low-temperature Rheology Test: The procedure suggested by Sui et al. [35] was followed to accomplish this test with a 4 mm parallel plate geometry using a DSR. A master curve was created using this procedure at a low-temperature PG + 10 °C (−18 °C); the slope (mr) and relaxation modulus G (t) at 60 s were then estimated using the master curve. Next, a comparison was made between these values and the Superpave PG binder parameters for a 4 mm DSR [27,36]. A minimum (mr) value of 0.3 and a maximum G (t) value of 300 MPa are advised per the standards. A binder that performs well at low temperatures has a greater mr and a lower G (t).

### 2.5. Mix Design, Specimen Preparation, and Tests

The mix design used for Highway 32 was adopted for this study. The PG 58–28 virgin binder was used to prepare the control HMA. The RAP aggregate was treated as a stockpile with the virgin aggregates and was first split into two proportions on the 4.75 mm (No. 4) sieve. The RAP material’s binder content, 7.5% in this case, was estimated using the ignition method following the AASHTO T308-16 procedure [37]. The +4.75 mm and −4.75 mm RAP material fractions were selected to meet the Superpave gradation specifications. The same gradation was used for the control and bio-oil-modified HMA to determine the binder effect. To account for 85% of the RAP binder and 15% of the bio-modifiers, the total percentage of the RAP was 72.5%, based on the aggregate weight used. The remaining weight was redistributed to the other aggregates using the job mix formula. The same gradation and mix designs were used to prepare the bio-oil-modified mixes with WWS. The mix design and gradation of the HMA are displayed in Table 3 and Table 4.

### 2.6. Mixing and Preparation

The virgin aggregates were heated at 163 °C (325 °F) overnight. The RAP binder was heated separately at 110 °C (230 °F) for 1 h to make it workable. The bio-modifiers and binder were blended and heated at 132 °C (270 °F) for 1 h. These heated materials were mixed in the laboratory, maintaining a standard mixing temperature of 149 °C (300 °F). Each batch was short-term aged for 2 h after mixing at the compaction temperature of 143 °C (290 °F). After two hours, a pre-weighed amount of mix was poured into a mold and compacted using the Superpave Gyratory Compactor (SGC). A cylindrical sample with a 150 mm diameter and a 75 mm thickness was compacted for the rutting resistance performance test. A 150 mm diameter and 100 mm height samples were compacted initially for the fatigue and low-temperature cracking resistance performance test. They were further resized to a 50 mm thickness according to specification requirements. The volumetric properties of the specimens were tested to conform to the 7 ± 0.5% air void requirement.

The same bio-oil modified mix was replicated by using 1%WWS in addition to other binders. The WWS was kept at a room temperature of 25 °C and added to the bio-modifiers before mixing with the aggregate. The compaction was completed at 116 °C (240 °F), 10 °C (50 °F) lower than the HMA compaction temperature. Only the fatigue and low-temperature performance tests were conducted on the mixes with WWS since rutting is not a significant concern on high RAP content HMA mixes.

### 2.7. Mix Testing

Rutting Resistance Test using Asphalt Pavement Analyzer (APA): A rutting resistance test on a dry-conditioned 75 mm specimen using an APA was conducted following the AASHTO T 340 specification [38]. The samples were conditioned and tested using the virgin high-temperature grade of the binder (58 °C). The testing sample was subjected to a pressure of 69 kPa (100 psi) for 8000 cycles. A rut depth of up to 7 mm ± 2 is permissible for moderate traffic levels with 0.3 to <3 million design Equivalent Single Axle Loads (ESALs), following the APA performance specification for North Dakota roadways [39].

Fatigue Cracking Resistance Test using Semi-circular Bend (SCB): The fatigue cracking resistance of the mix was assessed using the SCB test in accordance with the Illinois-Flexibility Index Tester (I-FIT) procedure. A 50 ± 2 mm sample was examined at 25 °C after conditioning for 2 ± 0.2 h. I-FIT software v 4.1 was used to post-process the data and determine the Flexibility Index (FI) and fracture energy. The cut-off values for poor, intermediate, and good mix were identified as (less than 2.0), (2.0 to 6.0), and (more than 6.0), respectively [26].

Low-temperature Cracking Resistance Test using Disk-Shaped Compact Tension (DCT): The DCT test was employed to determine the specimens’ resistance to low-temperature cracking. This test was carried out in accordance with ASTM D7313 guidelines [40]. Low-temperature cracking resistance was determined by measuring the 50 ± 2 mm specimens’ fracture energy (Gf). After conditioning for eight hours, the specimens were evaluated at the binder’s low-temperature PG + 10 °C (−18 °C). A steady pace of 0.017 mm/s for Crack Mouth Opening Displacement (CMOD) was kept throughout the test. It is recommended that threshold fracture energy values of 400, 460, and 690 J/m^2^ be used for low, middle, and high traffic levels, respectively, in cold climates like North Dakota [41,42,43,44].

## 3. Results and Discussions

### 3.1. Effect of Bio-Oils on Binder Rheology

Figure 2a displays the bio-oil modified and virgin unaged binders’ G*/(sin δ) value. The 15% SO and 15% WCO modified binders demonstrated lower values than the AASHTO M320 specification value of 1 kPa [27]. Replacing 2.5% of the two bio-oils with the virgin PG 58–28 binder improved the unaged G*/sin (δ) value and maintained the RTFO high-temperature continuous PG of 58 °C. Figure 2b depicts the continuous PG of all binders. Four RTFO-aged samples of each binder were tested. All modified binders reduced the continuous high-temperature PG of the RAP from 82 °C to 58 °C.

Rutting Performance: Figure 3a depicts the MSCR J_nr-3.2_ values of the binders. The results indicate that all modified binders satisfied the J_nr-3.2_ AASHTO M332 [31] requirement for standard traffic. All bio-modified binders resulted in a higher-temperature rutting performance than the virgin binder. The 15SO and 15WCO modified RAP binders had an approximately 5% lower J_nr_ value than the control binder, whereas the 12.5SO_2.5V and 12.5WCO_2.5V modified samples had an approximately 25% lower J_nr-3.2_ value than the control binder. This outcome could be due to the relative softening potential of the bio-oils compared to the virgin binder. The SO binders have greater softening potential than WCO at high temperatures. The percentages of recovery results in Figure 3b indicate that all modified binders have an approximately 60% higher recovery potential than the virgin binder. The SO-modified RAP binders have an approximately 7% higher recovery potential than WCO-modified RAP binders. These results indicate that the SO binder performed better at high temperatures than WCO in softening and improving the modified RAP binder’s elastic nature. These results suggest that the bio-oil-virgin-modified RAP binders had better rutting resistance than the virgin and bio-oil-modified RAP binders without a virgin binder.

Fatigue Cracking Performance: The LAS test was conducted on the PAV-aged modified and control binders at 19 °C. Table 5 illustrates that the SO-modified binders had higher “A” values than the WCO-modified binders. The 15SO sample had 80% more “A” values than the virgin binders and approximately 10% higher “A” values than all other modified binders. The “B” parameter indicated that all modified binders had lower fatigue cracking performance than the virgin binder. The 15SO and 15WCO samples had nearly the same “B” values and were approximately 7% higher than the corresponding 12.5SO_2.5V and 12.5WCO_2.5V. A 2.5% SO virgin binder replacement with the virgin binder alone reducing the number of cycles to failure by 55%, while replacing the same amount of WCO resulted in only a 10% reduction. These results could be due to the SO’s relative softening potential than the WCO, which confirms the results completed by other researchers [18]. The number of cycles to failure results indicated that the 15SO modified binders had better fatigue cracking performance than all other binders. All bio-oil-modified binders performed better in fatigue cracking resistance than the virgin binder based on the number of cycles to failure. This indicates that bio-oils relax the RAP binder, improving fatigue performance.

Low-Temperature Cracking Performance: Relaxation moduli and master curve slope for different PAV-aged binders are displayed in Figure 4. This study did not complete the PAV aging of 100% RAP since the RAP binder was already aged. Both bio-oil modified RAP binders without a virgin binder (15SO and 15WCO) had a significantly lower G (t) of below 50 and an approximately 10% higher m_r_ value than the virgin binder. The SO-modified binders had a higher low-temperature cracking performance than the WCO-modified binders. These results could be due to SO’s relatively higher softening performance than WCO. Replacing 2.5% of both bio-oils by the virgin PG 58–28 binder resulted in a 36% reduction in the low-temperature m_r_ values. These results indicate that the bio-oils have better softening potential than the virgin binder, which confirms research performed in a previous study [18].

### 3.2. Effect of WWS on Binder Performance

Figure 5 displays the N_f_ result from the LAS fatigue cracking performance test and the G(t) and m_r_ results from the 4 mm parallel plate low-temperature cracking performance test. The effects of the different dosages of WWS binder performance were evaluated. The PG 58–28 binder was selected since the continuous PG grade of all modified binders was 58 °C. The results indicated that the 1%WWS, based on the weight of the total binder dosage, showed better fatigue and low-temperature cracking resistance values. The WWS caused the binder to foam significantly, which was expected due to the water content in the WWS [23]. The outcome shows that employing 1%WWS in conjunction with 15%SO, 15%WCO, 12.5SO_2.5V, and 12.5WCO_2.5V met all of the requirements for high RAP binder performance in terms of rutting, fatigue cracking, and low-temperature cracking, surpassing even that of PG 58–28 binders. In addition, the results indicated that WWS could potentially be used as a compaction aid.

### 3.3. Effect of Bio-Oils on HMA

All four selected modifiers, along with the virgin and 100% RAP, were selected to make the HMA based on the binder’s rheology results. The same mix design was replicated and compacted at 116 °C (240 °F) using 1% WWS on bio-oil-modified and control HMA, then tested for cracking performance.

Rutting Performance: A summary of the APA test results for all modified and control HMA tested at 58 °C is displayed in Figure 6a. Three samples were tested for each case. Test results indicated that the rut depths for all specimens were well below the failure criterion, 7 mm, used in the study. The maximum rut depth among the specimens is 5.19 mm, as observed for the virgin HMA specimens. The 15SO and 15WCO modified HMA exhibited an 8% and 25% lower rut depth, respectively, than the virgin HMA. The 12.5SO_2.5V and 12.5WCO_2.5V modified HMA registered a 42% and 37% lower rut depth, respectively, than the control mix. A 2.5% replacement of WCO by virgin binder demonstrated an approximately 15% lower rut depth than the same percentage replacement of SO by the virgin binder. These results could be due to the relative softening potential of SO compared to WCO. The WCO-modified HMA was found to have a better rutting performance than SO-modified HMA. Figure 6b illustrates the correlation between MSCR J_nr_3.2_ values of the binder test and the APA rut depth of the HMA. The MSCR test results correlate well with the APA rut depth, with an R^2^ value of 0.87. Generally, the result indicates that the impact of bio-oil on rutting shows a similar trend and correlation between binder and HMA, indicating bio-oils’ consistency in improving rutting performance.

Fatigue Cracking Performance: The fracture energy and Flexibility Index (FI) results are summarized in Figure 7. The results indicate that the 15SO and 12.5WCO_2.5V modified HMAs have good fatigue cracking performance with an FI closer to 4.5. The mixes exhibited 13% and 9% higher FI values than the virgin HMA. The fracture energy of the 15SO and 12.5WCO_2.5V modified HMA exhibited a 58% and 31% lower fracture energy, respectively, than the virgin HMA. Modified mixes of the 15WCO and 12.5SO_2.5V samples had lower FI values than the virgin HMA. The interaction between the SO and virgin binders did not help fatigue performance like the interaction between the WCO and virgin binder. These results could be due to the better-softening potential of SO than the virgin binder. The flexibility index of the 100%RAP value was less than one. The results indicate that 100% RAP is highly brittle and has the least fatigue cracking resistance. The bio-oils did not improve the fatigue cracking performance of the high RAP mix, which also confirms other researchers’ results [14]. This result indicates the bio-oils were insufficient to reduce the stiffness of high RAP mixes.

Low-Temperature Cracking Performance: Figure 8 summarizes the low-temperature fracture energy of all bio-oil-modified and control HMA. As expected, 100% of the RAP HMA displayed the least low-temperature fracture energy. Most modified HMA, except for the 12.5SO_2.5V HMA, resulted in higher low-temperature fracture energy than the control HMA and passed the minimum threshold fracture energy value for intermediate traffic (460 J/m^2^). These results confirm those obtained by other researchers [17,18]. The 15SO and 15WCO-modified HMAs have the same low-temperature cracking performance potential. These results demonstrate that the 12.5WCO_2.5V HMA resulted in a 12% higher low-temperature fracture energy than the 15WCO HMA. In contrast, the 12.5SO_2.5V HMA resulted in a 37% reduction in fracture energy compared to the 15SO HMA. These results could be due to the relative softening potential of SO compared to the virgin binder, as confirmed by other researchers [18]. Results show that bio-oils and small amounts of virgin binder enhance high RAP thermal cracking performance in cold climates.

### 3.4. Effect of WWS on Cracking Performance and Compaction Effort

Fatigue Cracking Performance: All bio-oil-modified HMAs were tested with 1% WWS by the total binder weight. The samples were compacted at 116 °C (240 °F). Figure 9a,b display the bio-oil-modified binders’ fracture energy and flexibility index with and without WWS. The results indicate a significant improvement in fracture energy and FI for WWS-modified mixes. The fracture energy increased by an average of 64% and 25% for the 15SO and 15WCO modified HMA, respectively. Using WWS on 12.5SO_2.5V modified HMA increased the fracture energy by 16%, whereas using WWS with 12.5WCO_2.5V modified HMA decreased the fracture energy by 20%. These results could be due to SO’s relatively higher softening nature than WCO. The FI value was increased for all WWS-modified HMA. The FI value improved by 7% and 50% for the 15SO and 12.5SO_2.5V modified HMA, respectively, whereas it was enhanced by 3% and 9% for the 15WCO and 12.5WCO_2.5V modified HMA, respectively. These results indicate that SO works better with WWS than WCO. This information also suggests that the shortcomings of SO with a virgin binder in terms of fatigue performance can be mitigated using WWS. Modification of the control HMA by WWS resulted in an increase of the FI value by 60%. These results indicate that WWS works better in the presence of the virgin binder. The FI value revealed that applying 1%WWS on the bio-oil-modified binder resulted in better fatigue cracking performance than the control mix, except for the 15WCO HMA. This indicates that a small dosage of WWS added to bio-oils can significantly improve fatigue cracking performance and durability of high RAP mix.

Low-temperature Cracking Performance: Figure 10 illustrates the low-temperature fracture energy results of WWS-modified HMA. The use of WWS increased the fracture energy of 15SO and 15WCO by an average of 9%, whereas the fracture energy increased by an average of 45% on the 12.5SO_2.5V, the 12.5WCO_2.5V, and the control mix. These results indicate that WWS performs better in the presence of a virgin binder, which is expected since the foaming behavior occurs at the binder. This result also shows that applying 1% WWS significantly increased the low-temperature performance of all modified binder HMA and virgin binder HMA. This indicates that a small dosage of WWS added to bio-oils can significantly improve a high RAP mix’s low-temperature cracking performance and durability.

Compaction Effort: Figure 11 presents the mix’s bulk specific gravity (G_mb_) with and without WWS. These results reveal that the BSG of the mix with WWS is higher than that of the mix without WWS, which indicates that adding WWS improves the mix’s compatibility. This confirms a similar trend as a Zeolite-based warm mix additive [45].

## 4. Conclusions

Based on the study, higher percentages of RAP (up to 85% binder base or 72.5% aggregate base) can improve the rutting, fatigue cracking, and low-temperature cracking performance compared to control HMA using 1%WWS with 15SO and 12.5WCO_2.5V as modifiers maintaining the specification limit. When used without a virgin binder, the SO has a better softening and rejuvenating performance than the WCO at high, intermediate, and low temperatures. Using WWS in mixes improves cracking performance and compaction effort at 10 °C (50 °F) lower compaction temperature than the control HMA. These results indicate that WWS is a potential compaction aid additive. Large-scale field tests under different environmental conditions are recommended to support the conclusion.

## Figures and Tables

**Figure 1 materials-17-04276-f001:**
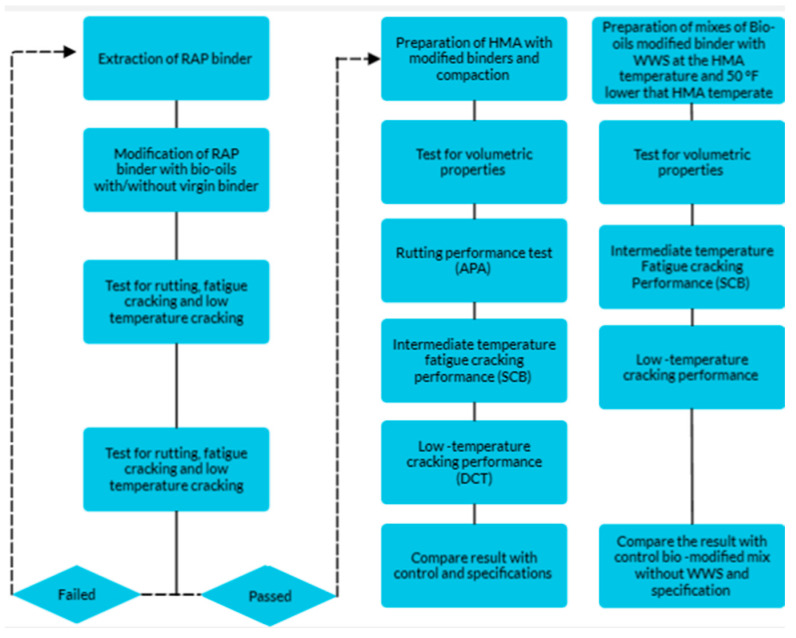
Experimental plan.

**Figure 2 materials-17-04276-f002:**
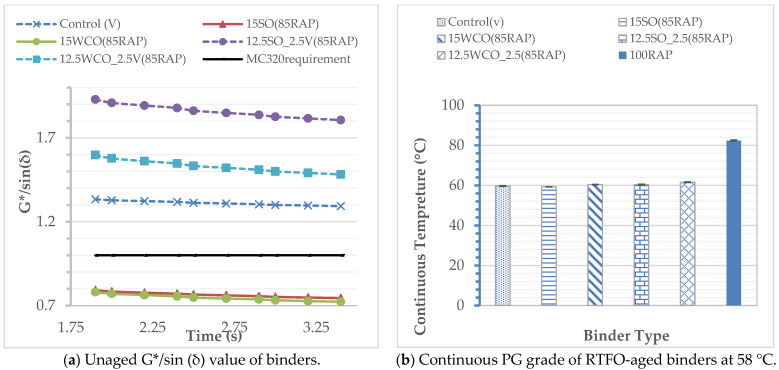
Modified binder rheology test results.

**Figure 3 materials-17-04276-f003:**
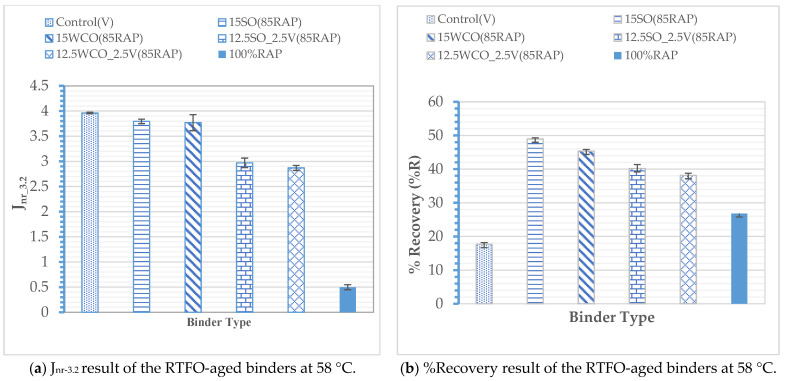
MSCR results.

**Figure 4 materials-17-04276-f004:**
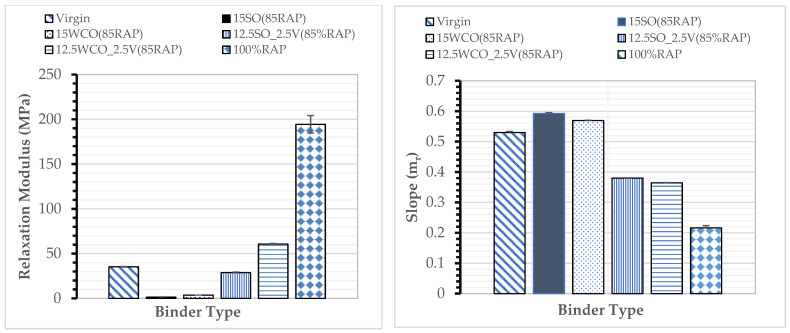
Relaxation modulus and slope (m_r_).

**Figure 5 materials-17-04276-f005:**
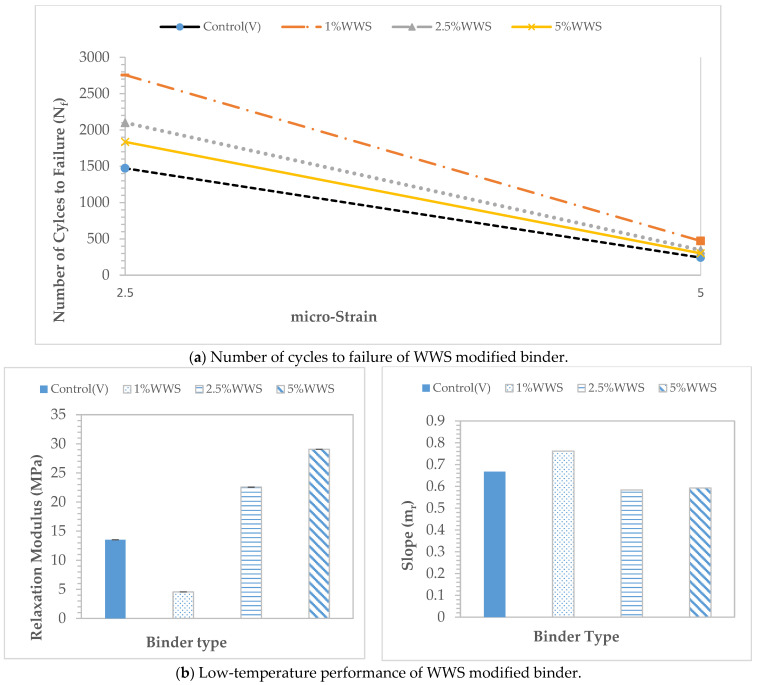
Effect of WWS on binder performance.

**Figure 6 materials-17-04276-f006:**
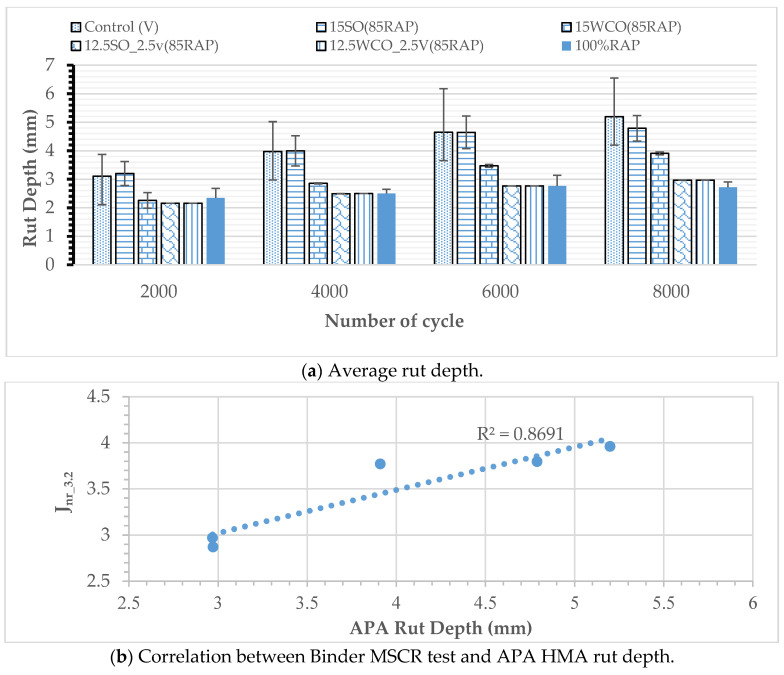
Effect of bio-oils on rutting performance.

**Figure 7 materials-17-04276-f007:**
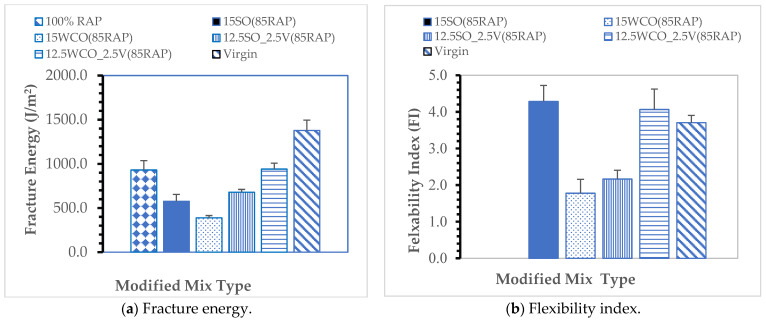
Fracture energy and flexibility index of bio-oil modified mixes.

**Figure 8 materials-17-04276-f008:**
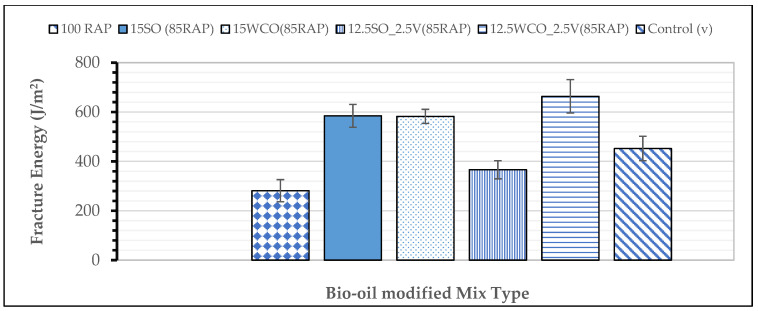
Low-temperature fracture energy of modified HMA.

**Figure 9 materials-17-04276-f009:**
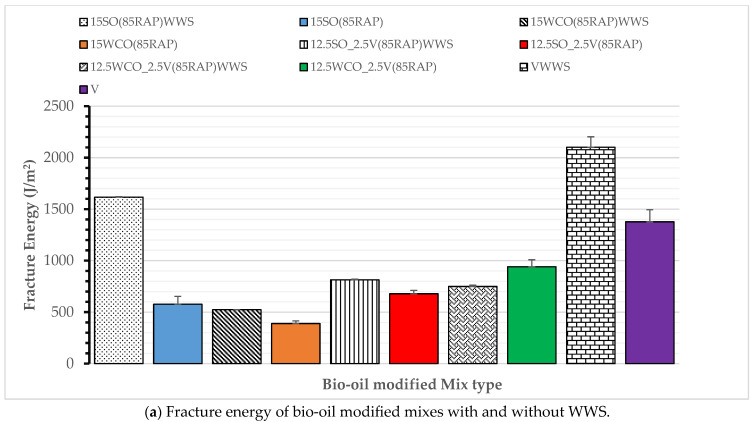
Effect of WWS on fatigue cracking performance.

**Figure 10 materials-17-04276-f010:**
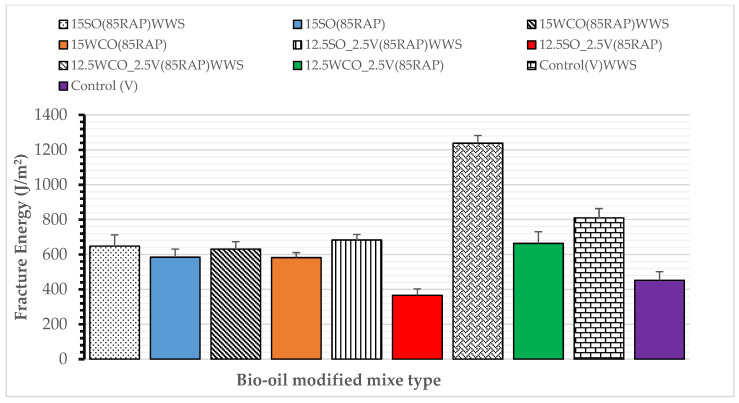
Effect of WWS low-temperature cracking performance.

**Figure 11 materials-17-04276-f011:**
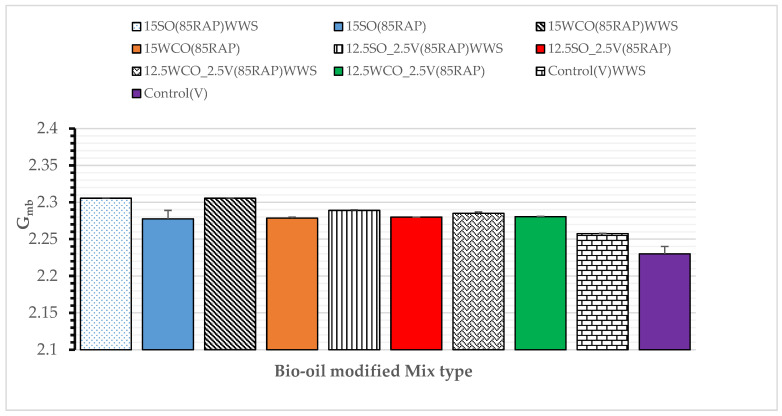
Effect of WWS on the bulk specific gravity of the mixes.

**Table 1 materials-17-04276-t001:** Properties of SO, WCO, and WWS.

Bio-Oil Type	SO	WCO	WWS
Viscosity (Stokes)	0.32–0.5	0.782	0.04–0.11
Specific Gravity at 25 °C	0.925	0.914–0.917	1–1.1
Acidic Value	0.1	0.1	0.1–0.2
Flash Point (°C)	260	220–230	250

**Table 2 materials-17-04276-t002:** Test on binders and mixes.

Types of Binders and Mixes	Test on Binders			Test on Mixes		
	Rutting (MSCR) at 58 °C	Fatigue Cracking (LAS) at 19 °C	Low-Temperature Cracking (4mmDSR) at −18 °C	APA at 58 °C	SCB at 25 °C	DCT at −18 °C
V	✓	✓	✓	✓	✓	✓
15SO	✓	✓	✓	✓	✓	✓
15WCO	✓	✓	✓	✓	✓	✓
12.5SO_2.5V	✓	✓	✓	✓	✓	✓
12.5WCO_2.5V	✓	✓	✓	✓	✓	✓
100R	✓	✓	✓	✓	✓	✓
15SO_(1%)WWS					✓	✓
V_ (1%)WWS		✓	✓		✓	✓
15WCO_ (1%)WWS					✓	✓
12.5SO_2.5V_ (1%)WWS					✓	✓
12.5WCO_2.5V_ (1%)WWS					✓	✓

**Table 3 materials-17-04276-t003:** Mix designs and proportions of the modified and control mixes.

Virgin PG 58–28 HMA		SO and WCO Modified HMA	
Materials	Percent (%)	Materials	Percent (%)
Optimum Binder content (OBC)	6.4	Optimum Binder content (OBC)	6.4
Bio-oil modifiers as a % of OBC	0	Bio-oil modifiers as a % of OBC	12.5/15
Virgin binder as %of (OBC)	100	Virgin binder as a % of OBC	2.5/0
Binder from the dry RAP aggregate as % of OBC	0	Binder from the RAP as % of OBC	85
Crushed Rock	29	Crushed Rock	8
Crushed Fines	37	Crushed Fines	10
Washed Dust	13	Washed Dust	3.5
Washed Sand	21	Washed Sand	6
RAP	0	RAP	72.5

**Table 4 materials-17-04276-t004:** Gradation of bio-oil modified with high RAP HMA.

Sieve	Bio-Oil Modified & Control HMA	Unmodified 100%RAP	Lower	Upper
	Gradation	Gradation	Control Pt	Control Pt
5/8”	100	100	100	100
1/2”	93.35	97.50	90	100
3/8”	81.03	87.65		
#4	59.44	68.41	40	70
#8	41.37	47.88		
#16	30.27	34.48		
#30	20.23	23.19	15	35
#50	11.19	13.89		
#100	6.32	9.01		
#200	3.48	5.58	2	7

**Table 5 materials-17-04276-t005:** LAS result of the modified binders at 19 °C.

Binder Type	Control (V)	15SO (85RAP)	15WCO (85RAP)	12.5SO_2.5V (85RAP)	12.5WCO_2.5 (85RAP)	100RAP
A. avg	12,353.3	70,966.7	39,766.7	31,442.8	39,248.4	30,407.6
St. D.	859.8	2263.8	5655.8	3943.5	4766.7	10,447.6
COV (%)	6.96	3.19	14.22	12.54	12.15	34.36
B_avg_.	−2.4760	−2.8873	−2.8570	−3.0137	−3.0606	−3.8520
St. D.	0.01	0.02	0.03	0.01	0.06	0.19
COV (%)	0.39	0.76	0.88	0.42	2.05	4.96
Nf__2.5avg_	1279	5052	2908	1985	2384	854
St. D.	100.05	56.36	474.06	228.13	344.25	127.58
COV (%)	7.82	1.12	16.30	11.50	14.44	14.93
Nf__5avg_	230	656	402	245	287	59
St. D.	19.47	52.16	72.00	26.27	49.05	0.62
COV (%)	8.47	7.95	17.91	10.70	17.08	1.06

## Data Availability

The original contributions presented in the study are included in the article, further inquiries can be directed to the corresponding author.

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
