# Peer review of "Effect of Bio-Oils and Wastewater Sludge on the Performance of Binders and Hot Mix Asphalt with High Reclaimed Asphalt Pavement Content"

_materials, 2024, doi:10.3390/ma17174276_

Round 1
Reviewer 1 Report
Comments and Suggestions for Authors
Dear authors,
your paper represents an interesting and important contribution aiming to introduce RAP as even more sustainable. Before publishing, I suggest the following amendments:
Lines 2-3: Please omit all abbreviations in the title and provide full names of terms instead.
Line 4: Omit the titles of authors, and leave just their names.
Lines 12: Provide full name for HMA. Also, check the rest of the text for the same issue.
Line 63: Erase the titles of chapters since these are parts of the Introduction.
Line 116: Move Figure 1 below the text where it was referred (now it is under line 90). Improve the quality of the figure, since the contrast is weak.
Lines 118-234: Provide references for the described methods as well as for the data presented in tables.
Lines 260-264: Charts are too big. Please rearrange them in a way to fit two in a row (like in line 307), enumerating them as (a) and (b), and use the same legend for both wherever it is justified.
Line 266 -: Replace „@“ with „at“ in the whole text.
Line 413-440: Present final remarks in conclusion as text and not in the form of bullets.
Line 434: Integrate the remark by the end of the Results and Discussion chapter, also as a part of the text – not in the form of bullets.
Final comment: The software for testing plagiarism has found 74% of matching to some other online documents and the most part is from a PhD thesis of one of the authors (up to 50%). Therefore, I propose to rewrite parts that are simply copied from the thesis. Even some whole tables are found to match… In this case, I advise you to present the data in another way – maybe as charts.
Author Response
The responses to the comments are attached.

Reviewer 2 Report
Comments and Suggestions for Authors
Comments:
In the review entitled « Effect of Bio-oils and Wastewater Sludge (WWS) on the Performance of Binders and HMA with High RAP Content" This review provides a thorough and detailed of The study demonstrates an innovative and promising approach to enhancing the performance of high RAP content asphalt mixtures by using bio-based rejuvenators and stabilizers, potentially paving the way for increased sustainability and efficiency in road construction. It is an essential field of study, and it has many issues to be studied, although the author has done excellent work and the paper is well-organized, the author should nonetheless take the following issues into account.1-The impact of red worms on treatment efficiency should be clarified and strategies to manage their presence should be included.
1- Summaries of results for some sections of the tests are not always clearly indicated. Add clear and concise summaries for each section of test results.
2-Some acronyms, such as HMA and RAP, are used without being defined upon their first appearance. Define all acronyms when they are first used in the document.
3- Some statements lack appropriate references. Add references for all important statements and data to strengthen the credibility of the article.
4- Some sentences are long and complex, making reading difficult. Revise the sentences to make them more concise and clear, avoiding overly long sentences.
Example "The results of the study indicate that the use of certain additives can significantly improve the performance of high RAP content mixtures by enhancing the durability and strength of the asphalt, which in turn can lead to longer-lasting roads and reduced maintenance costs."
Revised sentence: "The study results show that certain additives significantly improve high RAP mixtures' performance. This enhancement in durability and strength leads to longer-lasting roads and reduced maintenance costs."
5- Some tables, like Table 1, contain NA (Not Available) values. Complete the missing data or explain why they are not available.
6- What are the specific objectives of this study in terms of improving the performance of asphalt mixtures containing high RAP content?
7- Describe the three phases of the experimental plan used in this study. What are the key activities and tests conducted in each phase?
8- What types of binder tests were conducted to evaluate the properties of the modified binders, and what were the key findings?
Author Response
Responses to the comments are attached.

Reviewer 3 Report
Comments and Suggestions for Authors
The work presented is interesting, aiming at the sustainable use of waste or materials to create high quality asphalt floors. In general, it is well written and presented, which I recommend for publication after some improvements and modifications. In the summary the authors put the definitions of each abbreviation and RAP as soon as they put it in the introduction. The abbreviation HMA was never described in the document, no matter how common it may be in the area, it must be described. The diagram in Figure 1 should be more centered and the spaces between each segment should be well divided, as well as maintaining the same font size throughout the diagram. Line 121, is there any regulation followed to define this thickness for testing? Or how was this thickness defined? The presentation of the figures must be taken care of, for example, in figure 7 the figure a is extended vertically, and the figure b) is more or less square. In Figure 5, they are the same samples, just different tests, and the figures have different sizes and presentation forms. In general, homogenize the figures in size and presentation, text, legends... The work presented only refers to 30 references, which I believe is a very low number. They should make a more exhaustive comparison with literature, other materials that have been tested, and compare properties. Mostly the references are from the authors or that somewhat limit what was previously mentioned.
Author Response
Responses to the comments are attached.

Reviewer 4 Report
Comments and Suggestions for Authors
Title:
Ok
Abstract:
Review the acronyms, these should go after the first time the concepts to which they refer are mentioned, review RAP, HMA..., although it is suggested that this be used from the introduction, maintain the same criterion for the other concepts.
Keywords:
OK
Introduction:
Ok
Materials and methods
It is requested to support the adoption of the 15% dosage (mathematically, according to the tests carried out, references, or another method), in addition to specifying the units used of the 15%: volume?, weight?
The PG 58-28 binder is mentioned to be in common use in North Dakota, the binder specifications are requested so that the research can be repeatable.
It is recommended to place the brand and accuracy of the dynamic cutting rheometer used.
Line 124: check if you want to say only "virgin" and not "virgin binder"
Line 152: Specify the temperature or temperature range.
Line 177: It is suggested to place the proportions of the mixture design.
Line 204: specify the temperature or range of ambient temperature mentioned. It is suggested – if relevant – to place the relative humidity values of the environment in the same way
Line 205: a parenthesis needs to be closed so that the statement is understood.
Results and Discussions
Lines 281-282: when it is indicated that a better result was obtained, it is suggested to write the quantitative data, How much?
Line 290: show the results quantitatively – how smaller-
Imagery:
2, 3, 4, 5, 6a, 7, 8, 9, 10, 11: it is suggested to use the same criteria for the graphics (color, black/white or textures?, it is suggested to use colors since some textures are not displayed correctly as well as the same color/texture/or symbology adopted to represent the same compound if shown in different graphics)
Conclusions:
Ok
Author Response
Responses to the comments are attached.

Round 2
Reviewer 1 Report
Comments and Suggestions for Authors
Dear authors,
thank you for making the required corrections. After careful re-checking, I have found some minor issues that need to be improved.
- Some issues of aesthetic nature: There are many figures/charts and in some of them, legends are too big and even occupy more space than the chart itself (e.g. Figure 2). Therefore, I propose decreasing the font size of terms within the legends and if it is possible, rearranging terms in two columns to minimize occupied space. Moreover, words designating X and Y axes are bigger than the size of the text. Check please the instructions again about the size and font of text within the figures.
- As the 7. chapter you have added "Recommendations" - it is not usual practice in scientific papers, so I would kindly ask you to integrate the recommendations within the "Conclusion" by adding a few new sentences.
I have not detected serious language problems so far. In any case, and especially in parts that are going to be revised, use some English language software or the assistance of a professional lecturer.
Author Response
Dear authors,
thank you for making the required corrections. After careful re-checking, I have found some minor issues that need to be improved.
- Some issues of aesthetic nature: There are many figures/charts and in some of them, legends are too big and even occupy more space than the chart itself (e.g. Figure 2). Therefore, I propose decreasing the font size of terms within the legends and if it is possible, rearranging terms in two columns to minimize occupied space. Moreover, words designating X and Y axes are bigger than the size of the text. Check please the instructions again about the size and font of text within the figures.
- As the 7. chapter you have added "Recommendations" - it is not usual practice in scientific papers, so I would kindly ask you to integrate the recommendations within the "Conclusion" by adding a few new sentences.
All figures have been adjusted as per the comments.
Chapter 7, the recommendation part, has been merged with the conclusions.
Reviewer 3 Report
Comments and Suggestions for Authors
The authors addressed most of the comments but, however, figure 1 still has low image quality and the text boxes in columns 2 and 3 are still misaligned and with arrows to the right or left, instead of a simple arrow down. of equal size between each text box. In figure 2 the legend is larger than the graph, which should be the other way around. Figures 3 and 4, if 2 graphs are presented together in each figure, why not format equal graph size, equal reading font size as in figure 7???
Author Response
The authors addressed most of the comments but, however, figure 1 still has low image quality and the text boxes in columns 2 and 3 are still misaligned and with arrows to the right or left, instead of a simple arrow down. of equal size between each text box. In figure 2 the legend is larger than the graph, which should be the other way around. Figures 3 and 4, if 2 graphs are presented together in each figure, why not format equal graph size, equal reading font size as in figure 7???
All figures have been adjusted as per the comments.
Figure 1 has been changed and adjusted as per the comment.
Reviewer 4 Report
Comments and Suggestions for Authors
The authors have made the requested improvements. The work can be published.
Author Response
The authors have made the requested improvements. The work can be published.
Thank you.